



# Sensitivity of predicted ultrafine particle size distributions in Europe to different nucleation rate parameterizations using PMCAMx-UF v2.2

David Patoulias[1], Kalliopi Florou[1], Spyros N. Pandis[1,2]

[1] Institute of Chemical Engineering Sciences, Foundation for Research and Technology Hellas (FORTH/ICE-HT), Patras, Greece

[2] Department of Chemical Engineering, University of Patras, Patras, Greece

*Correspondence to*: Spyros N. Pandis (spyros@chemeng.upatras.gr)

**Abstract.** The three-dimensional chemical transport model, PMCAMx-UF v2.2, designed to simulate the ultrafine particle size distribution, was used to investigate the impact of varying nucleation mechanisms on the predicted aerosol number concentration in Europe. Two basic case scenarios were examined: the original ternary $H_2SO_4$-$NH_3$-$H_2O$ parameterization and a biogenic vapor-sulfuric acid parameterization. Using the organic-based parameterization, PMCAMx-UF predicted higher

$N_{10}$ (particle number above 10 nm) concentrations over Europe by 40-60% on average during the simulated period, which is a relatively small difference given the differences in the two assumed mechanisms. Adjusting the nucleation rate by an order of magnitude for both mechanisms led to an average change of ±30% in $N_{10}$ for the ternary ammonia case, and –30 to 40% for the biogenic vapor case. In the biogenic organic nucleation scenario, reducing the fresh nuclei diameter from 1.7 nm to 1 nm resulted in reductions in $N_{10}$ and $N_{100}$ by –13% and –1%, respectively. Incorporating extremely low-volatility organic

compounds (ELVOCs) as the nucleating species resulted in predicted increase in $N_{10}$ concentration by 10-40% over continental Europe compared to the ammonia parameterization. Model predictions were evaluated against field measurements from 26 stations across Europe during the summer of 2012. Among the tested scenarios, the measurements showed better agreement with the ternary ammonia and ELVOC-based parameterizations for $N_{10}$, whereas for $N_{100}$, all simulated cases appear to agree quite well with the field data.

## 1 Introduction

Aerosol nucleation together with direct emission from sources are the two principal processes for the introduction of new particles in the atmosphere. New particles formed by nucleation can either grow to larger sizes or can be lost by coagulation with existing particles (Kulmala et al., 2004; Merikanto et al., 2009; Pierce and Adams, 2009). New particle formation (NPF) through condensation of vapors (e.g., sulfuric acid, organics, ammonia, and nitric acid) is estimated to be responsible for up to

half of the global cloud condensation nuclei (CCN) and consequently affects considerably the cloud droplet number concentration (Adams and Seinfeld, 2002; Makkonen et al., 2009; Wang and Penner, 2009).



Various nucleation mechanisms have been proposed to describe the initial step of NPF. These mechanisms include sulfuric acid–water ($H_2SO_4$-$H_2O$) binary nucleation (Nilsson and Kulmala, 1998; Vehkamäki et al., 2002), sulfuric acid–ammonia–water ($H_2SO_4$-$NH_3$-$H_2O$) ternary nucleation (Bianchi et al., 2016; Kulmala et al., 2002; Napari et al., 2002; Yu, 2006), ion-induced nucleation (Jokinen et al., 2018; Kirkby et al., 2016; Laakso et al., 2002; Modgil et al., 2005), halogen oxide nucleation (Hoffmann et al., 2001), nucleation involving organic compounds (Li et al., 2019; Metzger et al., 2010; Weber et al., 2020), sulfuric acid–dimethylamine nucleation (Yao et al., 2018), and iodine oxides (Sipilä et al., 2016). The corresponding nucleation rates depend on the sulfuric acid vapor concentration, with numerous studies indicating a strong correlation between sulfuric acid levels and the rate of new particle formation (Kuang et al., 2008; Lee et al., 2019; Sihto et al., 2006).

While in sulfur-rich environments NPF can be often explained by a simplified acid-base model (Chen et al., 2012), model simulations (Anttila and Kerminen, 2003) and field measurements have showed that the condensation of sulfuric acid alone is often not enough to explain the observed growth rates of newly formed particles (Kuang et al., 2008). In environments with low sulfur dioxide levels new particle growth has been linked to organic vapors (Olenius et al., 2018; Yli-Juuti et al., 2020). To explain the growth of the fresh nuclei, condensation of organic species (Anttila and Kerminen, 2003) and heterogeneous reactions (Zhang and Wexler, 2002) have been proposed. Condensing low volatility organic vapors assist freshly formed particles to overcome the Kelvin effect growth barrier which appears for particles with diameters of a few nm (Semeniuk and Dastoor, 2018).

Organic aerosol (OA) is an important constituent of submicrometer particulate matter contributing more than 50% in many locations around the world (Reyes-Villegas et al., 2021; Ripoll et al., 2015). Secondary organic aerosol (SOA) is formed during the oxidation of both biogenic and anthropogenic volatile organic compounds (VOCs) and often accounts for most of the submicrometer OA (Hallquist et al., 2009; Jimenez et al., 2009; Schulze et al., 2017). VOCs of biogenic origin include terpenes such as isoprene ($C_5H_8$), monoterpenes ($C_{10}H_{16}$) and sesquiterpenes ($C_{15}H_{24}$) (Curci et al., 2009; Vermeuel et al., 2023). The oxidation of terpenes leads to highly oxygenated organic molecules (HOMs) that can participate in NPF and contribute to the growth of pre-existing particles (Ehn et al., 2014; Jokinen et al., 2015; Weber et al., 2020).

Chemical transport models integrate our understanding of atmospheric processes and combined with atmospheric measurements can help us evaluate if this understanding is satisfactory. There have been a number of efforts to simulate ultrafine particle number concentration and NPF from ground-level and airborne observations (Leinonen et al., 2022; Lupascu et al., 2015; Matsui et al., 2013). PMCAMx-UF is a three-dimensional regional chemical transport model (CTM) developed by Jung et al. (2010) specifically for simulating ultrafine particles. Baranizadeh et al. (2016) updated the nucleation parameterization in PMCAMx-UF by integrating the Atmospheric Cluster Dynamics Code, which is based on quantum chemical input data. The observed number concentrations of particles larger than 4 nm could be reproduced within one order of magnitude for Europe at that stage showing that there is room for improvement. Fountoukis et al. (2012) performed simulations over Europe and compared the model predictions against size distribution measurements from seven areas. The model successfully reproduced hourly number concentrations of particles larger than 10 nm ($N_{10}$) within a factor of two for





more than 70% of the time. However, it regularly underpredicted the concentrations of particles larger than 100 nm ($N_{100}$) by 50%. Notably, these early versions of the model did not account for SOA condensation on ultrafine particles. Patoulias et al. (2015) addressed this limitation by incorporating the condensation of organic vapors on nanoparticles through the development of a new aerosol dynamic model, DMANx (Dynamic Model for Aerosol Nucleation extended), demonstrating its significant

impact on NPF. Julin et al. (2018) further extended the model by including the effects of amines on NPF and projected future changes in ultrafine particle emissions across Europe. The impact of secondary semi-volatile organic vapors on particle number concentrations was examined by integrating the volatility basis set (VBS) approach into PMCAMx-UF and applying the model over Europe (Patoulias et al., 2018). Including the VBS enabled the model to reproduce $N_{10}$ and $N_{100}$ ground measurements within a factor of two for 65% and 70% of observations, respectively. The model was further enhanced to incorporate multiple

generations of intermediate-volatility organic compounds (IVOCs) gas-phase oxidation, along with the formation and dynamic condensation of extremely low-volatility organic compounds (ELVOCs) from monoterpenes (Patoulias and Pandis, 2022).

Different nucleation parameterizations are used by global air quality and regional chemical transport models that present different parameter sensitivity. Riccobono et al. (2014) developed an empirical parameterization based on field measurements to describe the dependence of nucleation rates on sulfuric acid and oxidized biogenic compounds

concentrations. Kirkby et al. (2016) found that highly oxidized organic compounds play a role in atmospheric particle nucleation comparable to that of sulfuric acid. Gordon et al. (2016) simulated the monoterpene HOMs formation using an empirical yield of HOMs during the oxidation of monoterpenes. Sartelet et al. (2022) simulated the heteromolecular nucleation of extremely low-volatility organic compounds (ELVOCs) from monoterpenes and sulfuric acid and reported improved predictive ability for suburban sites during the summer.

The parameterizations of nucleation often involve adjusting the absolute nucleation rate with a nucleation tuner while maintaining its dependence on the concentrations of the participating vapors (Jung et al., 2010). Another important parameter is the initial nuclei diameter, that is the size of newly formed particles. Paasonen et al. (2018) investigated particle growth in a boreal forest, highlighting the model's sensitivity to initial nuclei diameter variations, which substantially impacted growth dynamics and subsequent cloud condensation nuclei (CCN) formation.

In this study, we explore the impact of ammonia and organic vapor-based nucleation parameterizations on predicted particle number concentrations (e.g. $N_{10}$ and $N_{100}$) and evaluate potential changes in model performance. Specifically, we investigate the effects of a) altering the nucleation rate, by an order of magnitude (both increase and decrease), b) modifying the nuclei diameter, and c) incorporating extremely low-volatility organic compounds (ELVOCs). Ground-level measurements from 26 European stations during the simulated period are used to evaluate PMCAMx-UF for the different used

parameterizations.





## 2 Model Description

The three-dimensional chemical transport model PMCAMx-UF simulates both the chemically resolved mass distributions and particle number distributions down to the nanometer size range (Fountoukis et al., 2012; Jung et al., 2010; Patoulias & Pandis, 2022; Patoulias et al., 2018). PMCAMx-UF is based on the PMCAMx (Gaydos et al., 2007) air quality model that describes 100 the processes of horizontal and vertical dispersion and advection, emissions, dry and wet deposition, aerosol dynamics and thermodynamics, aqueous and aerosol phase chemistry. The simulation of the aerosol microphysics, is handled in PMCAMx-UF by the updated version of the Dynamic Model for Aerosol Nucleation (DMANx), which simulates condensation, evaporation, new particle formation (NPF), and coagulation assuming an internally mixed aerosol (Patoulias et al., 2015). DMANx is based on the Two-Moment Aerosol Sectional (TOMAS) algorithm which tracks independently both the aerosol 105 number and mass distributions for each of the 41 logarithmically-spaced size bins between 0.8 nm and 10 μm (Adams and Seinfeld, 2002). In each bin, the particle density is calculated and updated continuously as a function of the corresponding composition. Each successive size bin boundary has twice the mass of the previous one to simplify the simulation of coagulation. The lowest boundary is at $3.75 \times 10^{-25}$ kg of dry aerosol mass per particle, corresponding to a dry diameter of 0.8 nm. The modelled particle components include ammonium, sulfate, nitrate, chloride, sodium, water, crustal material, elemental 110 carbon, primary organic aerosol (POA), and eight surrogate SOA components.

In the current study, the base case nucleation rate was computed using a ternary $H_2SO_4$-$NH_3$-$H_2O$ parameterization assuming a scaling factor of $10^{-7}$ (Fountoukis et al., 2012; Napari et al., 2002). For $NH_3$ concentrations below the threshold value of 0.01 ppt, the binary $H_2SO_4$-$H_2O$ parameterization of Vehkamäki et al. (2002) was used. Coagulation is both an important sink of aerosol number in the atmosphere, but also a mechanism by which freshly nucleated particles grow to larger 115 sizes. Following Adams and Seinfeld (2002), the effects of gravitational settling and turbulence on coagulation are assumed negligible and particles coagulate predominantly via Brownian diffusion. The coagulation coefficients were calculated based on the wet diameters of the particles, which were determined following the method of Gaydos et al. (2005). For smaller particles, the corrections of Dahneke (1983) for non-continuum effects were used. The coagulation algorithm uses an adaptive time step, which does not allow increase in aerosol number or mass concentration in any size bin by more than an order of 120 magnitude or a decrease by more than 25% in each step.

During the last years, PMCAMx-UF has been extended to include chemical aging of semi-volatile anthropogenic organic vapors, oxidation of intermediate-volatility organic compounds (IVOCs), and the production of extremely low-volatility organic compounds (ELVOCs) by monoterpenes (Patoulias and Pandis, 2022). Additional information describing the evolution and evaluation of PMCAMx-UF model can be found in previous publications (Fountoukis et al., 2012; Jung et 125 al., 2010; Patoulias et al., 2018; Patoulias and Pandis, 2022).

The extended Statewide Air Pollution Research Center (SAPRC) gas phase chemical mechanism is used in PMCAMx-UF (Carter, 2000; Environ, 2005). SAPRC contains 219 reactions of 64 gases and 18 free radicals. The SAPRC version used for the current study includes five lumped alkanes (ALK1–5), two lumped aromatics (ARO1 and ARO2), two



lumped olefins (OLE1 and OLE2), a lumped monoterpene (TERP), isoprene (ISOP), and a lumped sesquiterpene species
(SESQ).

A pseudo-steady-state approximation (PSSA) is used for the simulation of sulfuric acid vapor concentration. This allows a significant increase in computational speed with a minor loss in accuracy (Pierce and Adams, 2009). Condensation of ammonia on ultrafine particles is modelled following Jung et al. (2010) and ends when sulfate is entirely neutralized forming ammonium sulfate. The assumption that the system is always in equilibrium is used for the partitioning of nitric and
hydrochloric acids (as nitrate and chloride, respectively) to particles in the accumulation mode range in PMCAMx-UF. In this version of PMCAMx-UF, the water content of the organic aerosol is neglected, and the aerosol water is associated with the inorganic aerosol components.

## 2.1    Nucleation mechanisms

PMCAMx-UF has the option of using a number of nucleation parameterization (Baranizadeh et al., 2016; Fountoukis et al.,
2012). In this work, we investigate two types of parameterizations, a ternary $H_2SO_4$–$NH_3$–$H_2O$ parameterization (ammonia parameterization) and a second including the products of the biogenic VOC oxidation, the $H_2SO_4$–bSOA–$H_2O$ parameterization (bSOA parameterization). Several variations of these schemes are examined.

The ammonia parameterization is the default parameterization in PMCAMx-UF. In the base case, the selected nucleation tuner is equal to the value of $10^{-7}$. The fresh nuclei diameter $d_p$ ranges between 0.8 and 1.2 nm as a function of
ammonia, sulfuric acid, temperature, and RH (Napari et al., 2002). The parameterization is valid for temperatures between 240 and 300 K, relative humidity of 5-95%, ammonia mixing ratios of 0.1-100 ppt, sulfuric acid concentration of $10^4$-$10^9$ molecules cm$^{-3}$, and nucleation rates between $10^{-5}$ - $10^6$ cm$^{-3}$ s$^{-1}$ (Napari et al., 2002).

The participation of biogenic secondary organic compounds in the nucleation mechanism together with sulfuric acid is based on the semi-empirical parameterization by Riccobono et al. (2014):

$$J_{1.7} = k[BioOxOrg][H_2SO_4]^2, \tag{1}$$

where $J_{1.7}$ is the nucleation rate (in cm$^{-3}$ s$^{-1}$) for particles with mobility diameter equal to 1.7 nm, $k$ is a fitted parameter that was originally set equal to $3.27 \times 10^{-21}$ molecule$^{-3}$ cm$^6$ s$^{-1}$, [BioOxOrg] is the concentration of monoterpene oxidation products (in molecule cm$^{-3}$), and [$H_2SO_4$] is the concentration of sulfuric acid (in molecule cm$^{-3}$) in the atmosphere.

The above parameterization needs to be adjusted to be compatible with the VBS parameters. PMCAMx-UF lumps
all monoterpenes such as α-pinene, β-pinene, limonene, etc., into one surrogate species. The monoterpene atmospheric oxidation products, using the VBS, are represented by 4 surrogate species with effective volatility at 298 K, C*=1, 10, 100, and 1000 µg m$^{-3}$. We assume here that only the product with the lowest volatility (C* = 1 µg m$^{-3}$) participates in new particle formation. This species is used effectively as a surrogate for the compounds with much lower volatility participating in the process. The sensitivity of our results to this choice will be examined in a subsequent section. To calculate the corresponding
nucleation rate constant (instead of the value used in Equation 1), we used the available nucleation rate measurements summarized in the work of Chen et al. (2012) (Fig. S1). The diagonal lines in Chen et al. (2012) represent the maximum and



minimum boundary of atmospheric measurement. To get a zeroth order estimation of an appropriate rate constant value, the

predicted concentrations of sulfuric acid vapor and biogenic SOA ($C^* = 1$ µg m$^{-3}$) vapor during the PMCAMx-UF simulation

were used to calculate the nucleation rate constant using Equation 1. Least-square fitting of the predicted nucleation rate to the

average of the maximum and minimum boundaries of atmospheric measurements shown in Chen et al. (2012) yields a rate

constant ($k$) of 0.1 x 10$^{-21}$ molecule$^{-3}$ cm$^6$ s$^{-1}$ . This value was used in our PMCAMx-UF parameterization resulting in Eq. 2:

$$J_{1.7} = 1 \times 10^{-22} [bSOA_{C_1^*}][H_2SO_4]^2, \tag{2}$$

where $bSOA_{C_1^*}$ corresponds to the concentration of the biogenic secondary organic vapor from the oxidation of monoterpenes

with a saturation concentration ($C^*$) of 1 µg m$^{-3}$ at 298 K. The use of this surrogate VBS species instead of the BioOxOrg of

Riccobono et al. (2014) results in different rate constant.

## 2.2    Description of sensitivity tests

A series of sensitivity tests have been performed for the ammonia and biogenic organic parameterizations described above

(Table 1). To evaluate the impact of the absolute nucleation rates, we increased the rate constant by an order of magnitude for

both the ammonia and bSOA parameterizations in Cases 2 and 5. Similarly, the rate constant was also decreased by an order

of magnitude in Cases 3 and 6. For the organic nucleation scenario two additional cases have been investigated. In Case 7, the

initial nuclei diameter was reduced from 1.7 nm to 1 nm.

**Table 1. Nucleation parameterization scenarios.**

|  | Case 1 | Case 2 | Case 3 | Case 4 | Case 5 | Case 6 | Case 7 | Case 8 |
|---|---|---|---|---|---|---|---|---|
| Third Species | Ammonia | Ammonia | Ammonia | Organic | Organic | Organic | Organic | Organic |
| $C^*$ (µg m$^{-3}$) |  |  |  | 1 | 1 | 1 | 1 | $10^{-5}$ |
| $k$ (molecule$^{-3}$ cm$^6$ s$^{-1}$) | $10^{-7\,*}$ | $10^{-6\,*}$ | $10^{-8\,*}$ | $10^{-22}$ | $10^{-21}$ | $10^{-23}$ | $10^{-22}$ | $10^{-21}$ |
| Particle diameter (nm) | 0.8 - 1.2 | 0.8 - 1.2 | 0.8 - 1.2 | 1.7 | 1.7 | 1.7 | 1 | 1.7 |

[*] nucleation tuner for ternary ammonia nucleation (dimensionless)

In Case 8, extremely low-volatility organic compounds (ELVOCs) with a saturation concentration ($C^*$) of $10^{-5}$ µg m$^{-3}$ were

introduced as the organic component in the nucleation mechanism. The ELVOCs are assumed to be produced by the oxidation

of monoterpenes with a yield of 5% (Patoulias and Pandis, 2022; Rissanen et al., 2014), and a rate constant ($k$) of 1.0 x 10$^{-21}$

molecule$^{-3}$ cm$^6$ s$^{-1}$, with the corresponding nucleation rate being calculated by:

$$J_{1.7} = 1 \times 10^{-21} \left[ bSOA_{C_{10^{-5}}^*} \right] [H_2SO_4]^2, \tag{3}$$





## 3 Model Application

The modelling domain of PMCAMx-UF in this application covers a 5400 × 5832 km$^2$ region in Europe, with a grid resolution of 36 × 36 km and fourteen vertical layers spreading up to 7.5 km. The modelling period focuses on the PEGASOS campaign and includes a total of 34 days in 2012, starting on June 5 until July 8 of 2012.

A rotated polar stereographic map projection was used for the simulations by PMCAMx-UF. To minimize the effect of the initial conditions on the results, the first two days of each simulation were excluded from the analysis. Relatively low

and constant values have been used for the boundary conditions allowing the predicted particle number concentrations over central Europe to be determined by the emissions and corresponding processes simulated by the model. The boundary conditions and their effects on the predicted number concentrations by PMCAMx-UF in this domain are discussed in previous publications (Patoulias et al., 2018; Patoulias and Pandis, 2022).

Meteorological inputs to PMCAMx-UF include temperature, pressure, horizontal wind components, water vapor,

vertical diffusivity, clouds, and rainfall. The above inputs were generated by the Weather Research and Forecasting (WRF) model (Skamarock et al., 2005). WRF was driven by geographical and dynamic meteorological data generated by the Global Forecast System (GFSv15) of the National Oceanic and Atmospheric Administration/National Centers for Environmental Prediction. The layers of WRF and PMCAMx-UF were aligned with each other. The WRF simulation was periodically re-initialized every 3 days with observed conditions to ensure accuracy in the corresponding fields used as inputs in PMCAMx-

UF. Each field was provided with fidelity appropriate to the chosen grid resolution of the model as the measurements were pre-processed by the WPS (WRF Preprocessing System) package.

The particle emissions were based on the pan-European anthropogenic particle number emission inventory and the carbonaceous aerosol inventory (Kulmala et al., 2011) developed during the European Integrated project on Aerosol, Cloud, Climate, and Air Quality Interactions (EUCAARI) project. The resulting number and mass inventories contain both number

emissions and consistent size-resolved composition for particles over the size range of approximately 10 nm to 10 μm.

### 3.1    Measurements

The model results were compared against measurements in 26 ground sites, which are available in the European Supersites for Atmospheric Aerosol Research (EUSAAR), and EBAS databases (https://ebas.nilu.no) and the Aerosols, Clouds and Trace gases Research Infrastructure (ACTRIS) (https://actris.nilu.no). Particle size distribution measurements at all sites were made

using either a Differential Mobility Particle Sizer (DMPS) or a Scanning Mobility Particle Sizer (SMPS). Information about all the measurement stations can be found in Table S1.



## 4 Results

### 4.1     Base case ammonia and organic parameterizations

The average ground level number concentrations for both base case nucleation parameterizations are shown in Fig. 1. For the
ammonia parameterization, the $N_{tot}$ and $N_{10}$ have the highest concentrations in the Iberian Peninsula, Netherlands, Poland, and
Turkey due to nucleation. For $N_{50}$ and $N_{100}$ the highest concentrations are predicted in the Balkans and the Mediterranean Sea
due to the high emissions of sulfur dioxide in the surrounding areas and the intense photochemistry. High $N_{50}$ and $N_{100}$ values
are also predicted in Poland, Russia, and Ukraine due to urban and industrial emissions. Nucleation is predicted to increase the
total average number concentration by 160%. For $N_{10}$ and $N_{100}$ the enhancement due to nucleation was 140% and 45%,
respectively. The predicted ammonia concentration exceeded 8 ppb in Germany, the Netherlands, France, northern Italy,
Poland, and Russia, as shown in Fig. 2a, primarily due to intensive agricultural activities in these regions. Fig. 2b presents the
average sulfuric acid concentration, which, unlike ammonia, was higher over marine areas such as the Mediterranean Sea and
particularly the Aegean Sea, as well as coastal regions of the Atlantic Ocean including the Portuguese, Spanish, and French
coasts. These elevated levels are attributed to significant SO₂ emissions from maritime shipping activities and high OH levels
in these high relative humidity sunny regions. Fig. 2c depicts the average predicted nucleation rate, with values exceeding 1
cm$^{-3}$ s$^{-1}$ in parts of Portugal, northern Spain, the United Kingdom, the Balkans, Turkey, Poland, and Russia. In contrast, the
average nucleation rate across the remainder of Europe generally remained below 0.2 cm$^{-3}$ s$^{-1}$.

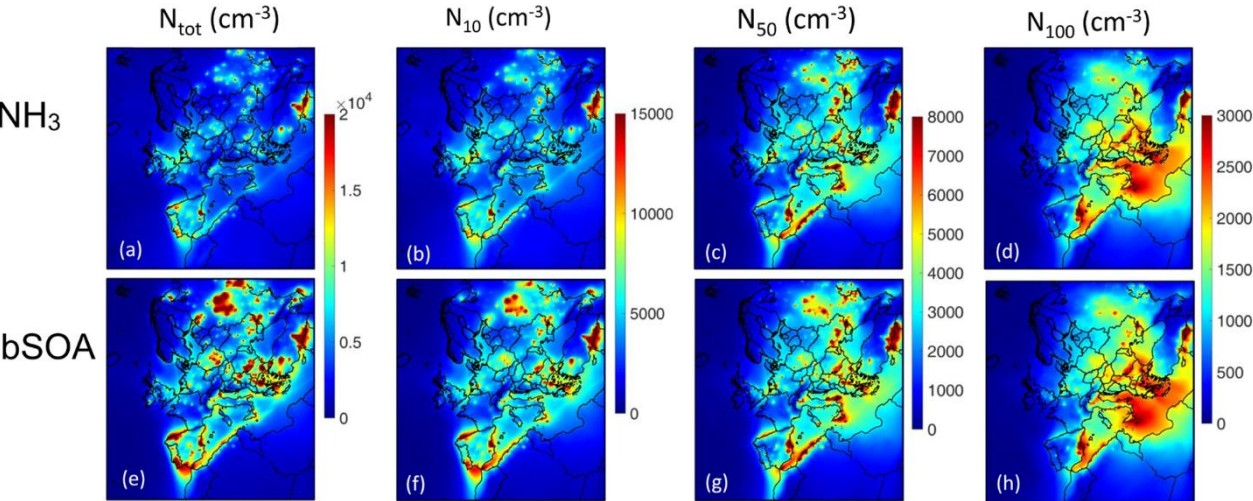

**Figure 1: Average ground level number concentrations (in cm$^{-3}$) for the ternary ammonia nucleation simulation during 5 June-8**
**July 2012 for: (a) all particles ($N_{tot}$); and particles above (b) 10 nm ($N_{10}$); (c) 50 nm ($N_{50}$); and (d) 100 nm ($N_{100}$). Average ground**
**level number concentrations (in cm$^{-3}$) for the biogenic semi-volatility organic nucleation simulation during 5 June-8 July 2012 for:**
**(e) all particles ($N_{tot}$); and particles above (f) 10 nm ($N_{10}$); (g) 50 nm ($N_{50}$); and (h) 100 nm ($N_{100}$). Different scales are used.**





For the biogenic parameterization the predicted $N_{tot}$ and $N_{10}$ have the same spatial patterns as with the ammonia parameterization, but with higher predicted levels especially in Italy, Russia, the Balkans, and parts of the Mediterranean Sea (Fig. 1e-f). The highest predicted concentrations of $N_{50}$ and $N_{100}$ are almost identical with those predicted by the ammonia parameterization (Fig. 1c-d). When these predictions were compared to the no-nucleation scenario, the enhancement attributable to nucleation in this simulation was approximately 300% for $N_{tot}$, 180% for $N_{10}$ and 50% for $N_{100}$. The gas phase

concentration of the bSOA component with $C^*=1$ μg m$^{-3}$ was predicted to be elevated in forested regions of central and northern Europe, including the Scandinavian countries, northern Russia, and Georgia (Fig. 2d). The average sulfuric acid concentration remained similar to that in the previous case. Incorporating bSOA ($C^*=1$ μg m$^{-3}$) as a third species resulted in an increased nucleation rate, with higher average values (above 4 cm$^{-3}$ s$^{-1}$) in regions such as Portugal, northern Spain, the Mediterranean Sea, Greece and the Aegean Sea, the Balkans, Turkey, Poland, and Russia (Fig. 2f). Despite the relatively low

concentration of semi-volatile biogenic organics in the Mediterranean Sea region, the high concentrations of sulfuric acid resulted in an elevated predicted nucleation rate (Fig. 2e).

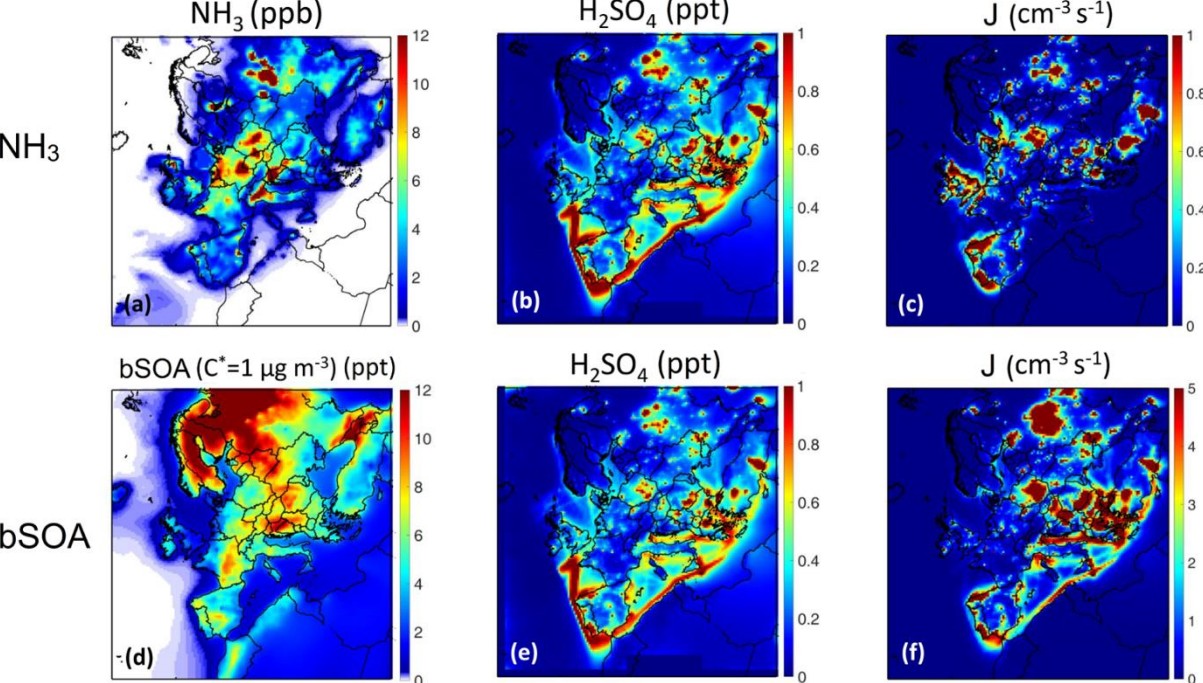

**Figure 2: Ground level average concentration of a) ammonia (NH₃) (in ppb) and (b) sulfuric acid (in ppt) and c) nucleation rate $J$ (in cm$^{-3}$ s$^{-1}$) for the ternary ammonia nucleation. Ground level average mass concentration of (d) biogenic semi-volatility secondary**
**organics compounds with $C^*=1$ μg m$^{-3}$ (in ppt) and (e) sulfuric acid (in ppt) and (f) nucleation rate $J$ (in cm$^{-3}$ s$^{-1}$) for the organic nucleation during 5 June-8 July. Different scales are used.**



The PMCAMx-UF number concentration predictions using the biogenic nucleation parameterization are higher than those predicted using the ammonia parameterization in most areas. More specifically, the predicted $N_{tot}$ is 80-150% higher,

and the $N_{10}$ 30-60% higher in regions with intense nucleation (Fig. 3a-b, e-f). On the other hand, $N_{tot}$ decreased by approximately 25% and $N_{10}$ by 10% in southern England, northern France, and the Netherlands when the organic parameterization replaced the ammonia one. $N_{50}$ increased by 10-30% in Greece and Russia, while it decreased by about 10% in the United Kingdom and Germany (Fig. 3g). The changes in $N_{100}$ were minor, ranging from 5-10% across the European domain (Fig. 3h).

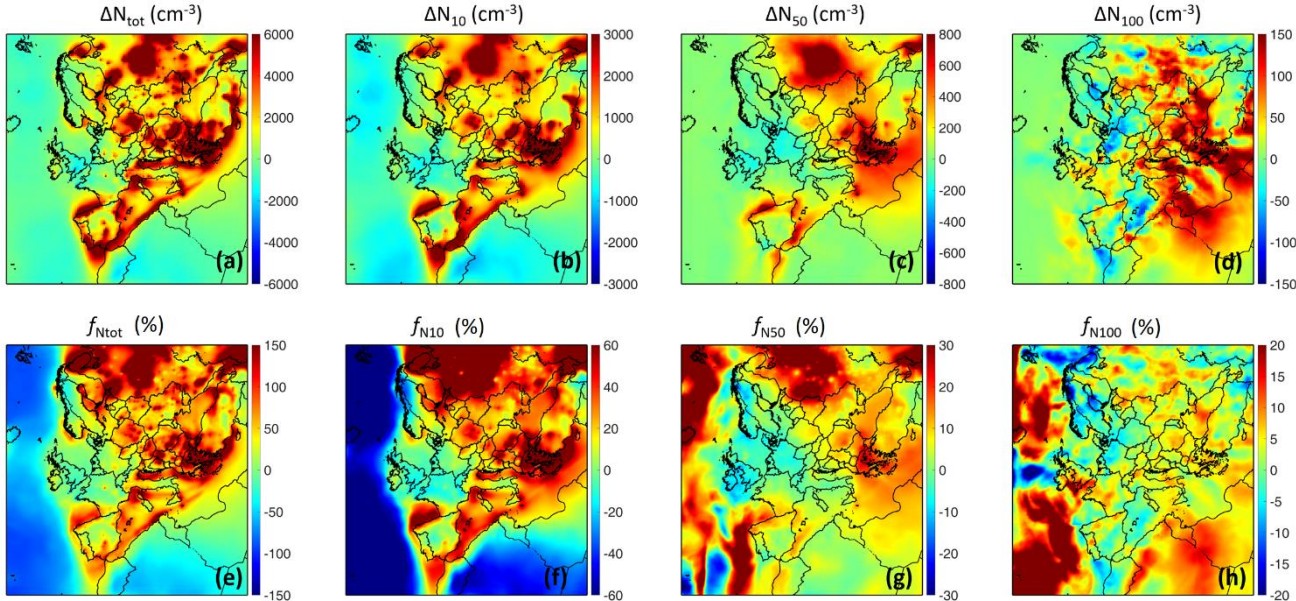


**Figure 3: Average ground change [biogenic-ammonia parameterization] of number concentration (in cm⁻³) (a-b-c-d) and fractional increase ($f_{Nx}$) of number concentration (in %) (e-f-g-h) during 5 June-8 July 2012 for: (a, e) all particles ($f_{Ntot}$); (b, f) particles above 10 nm ($f_{N10}$); (c, g) above 50 nm ($f_{N50}$); and (d, h) above 100 nm ($f_{N100}$). Different scales are used. Positive values indicate higher concentrations for the biogenic vapor case.**


## 4.2    Evaluation of the model

The predictions of the two simulations (ammonia and bSOA base case parameterizations) were compared against hourly $N_{10}$ and $N_{100}$ field measurements. The overall hourly normalized mean bias (NMB) for $N_{10}$ was found to be −16 % for ammonia and 2% for bSOA case, while the $N_{100}$ NMB was close to 7% for both cases (Fig. 4). The overall normalized mean error (NME)

for the $N_{10}$ was 54% in the ammonia case and 61% in the bSOA one. This indicates that the overall performance of the two parameterizations is comparably effective. Despite their inherent differences, both parameterizations demonstrated robust





performance, even when evaluated on an hourly basis. At 15 stations (ANB, CBW, DSN, DSW, HYY, ISP, KPU, MLP, PRG, USM, VSM, WLD, ZUG, PAT, SPC) (Table S1) the NMB for $N_{10}$ is lower in the case of the bSOA parameterization than in the case with ammonia. However, there are six stations (ASP, FNK, GDN, VRR, NEO, THE), for which the use of biogenic

organic nucleation significantly increases the $N_{10}$ NMB compared to the ammonia case (Fig. 4a). In both simulations, the $N_{10}$ NME remains below 60% for most of the stations, with a difference of less than 5% between the two cases. Notably, the predictions in six stations (ASP, FNK, GDN, VRR, NEO, THE) had significantly lower NME values for the ammonia mechanism (Fig. 4b). Of these, three stations are in Greece, one in Malta, one in Finland, and one in Sweden.

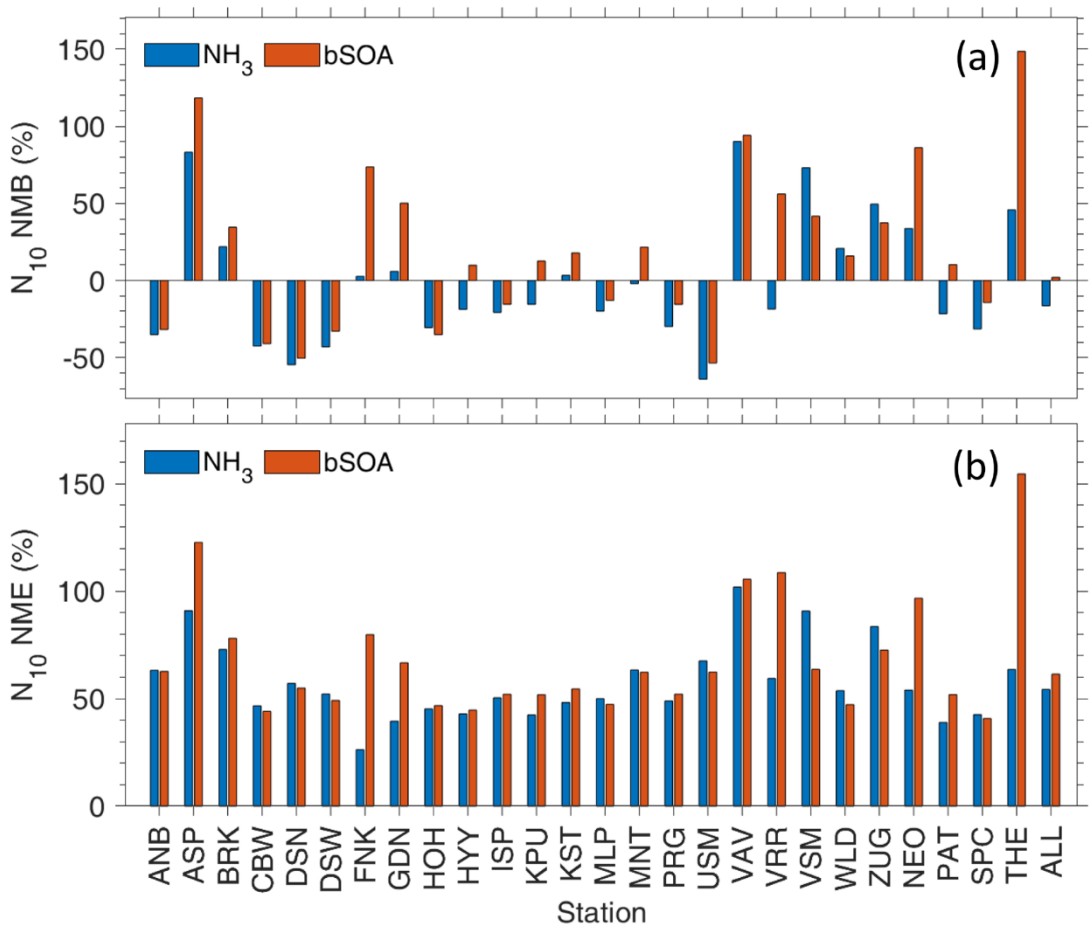


**Figure 4: The hourly (a) normalized mean bias (NMB) (in %) and (b) normalized mean error (NME) (in %) of $N_{10}$ for 26 stations. Blue bars are used for the simulation with ternary ammonia nucleation and red bars for the biogenic parameterization.**

For $N_{100}$ the NME for both cases was similar and equal to 48%. The hourly $N_{100}$ NMB for all stations ranged between

-40% and 80% (Fig. 5a). No significant differences (less than 10%) appear in the NMB of $N_{100}$ for the two simulations, with





the only exceptions those of ASP, FNK and THE stations. For FNK and THE stations in Greece, the ammonia parameterization shows less error, while the opposite is the case for ASP in Sweden (Fig. 5b).

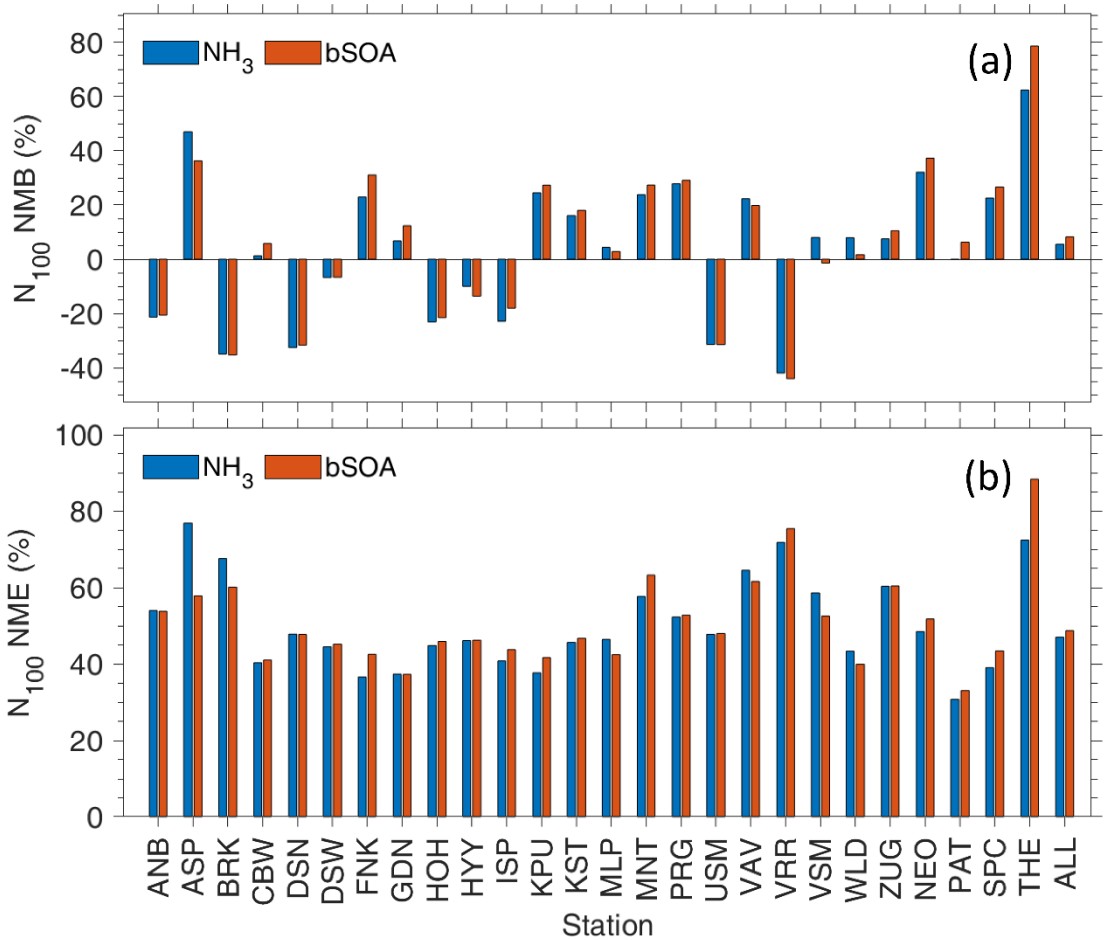

**Figure 5: The hourly (a) normalized mean bias (NMB) (in %) and (b) normalized mean error (NME) (in %) of $N_{100}$ for 26 stations.**
**Blue bars are used for the simulation with ternary ammonia nucleation and red bars for the biogenic parameterization.**

## 4.3 Results of sensitivity tests

### 4.3.1 Effect of scaling the ammonia and biogenic nucleation rate parameterizations

For the ammonia parameterization two additional cases were investigated (Case 2 and 3; Table 1). In Case 2 an increase of the
nucleation rate by a factor of 10 caused a 70-100% increase in $N_{tot}$ (4000-6000 cm$^{-3}$) and a 40-60% increase in $N_{10}$ (over 2000 cm$^{-3}$) in the regions with intense nucleation like the Iberian Peninsula, central Europe, the Balkans, and Turkey (Fig. S2). For $N_{50}$ an increase of about 10-20% (300-500 cm$^{-3}$) was predicted in the Balkans, eastern Mediterranean, Poland, and Russia. For



$N_{100}$ the change was small (less than 10%) with the most significant increase of 5%-8% in the Balkans, Eastern Mediterranean Sea, and Russia (Fig. S2).

In Case 3 a reduction by a factor of 10 in the nucleation rate resulted in an overall reduction in all investigated number concentrations for the modelled domain. A 40-60% reduction in $N_{tot}$ (about 2000-3000 cm$^{-3}$) and a 30-40% decrease in $N_{10}$ (over 15000 cm$^{-3}$) was predicted in the regions with intense nucleation (Fig. S3). The $N_{50}$ decreased about 15-20% mainly in the Balkans, Mediterranean, Eastern Europe, Turkey, and parts of Scandinavia. For $N_{100}$ there was a 5%-10% decrease in the Balkans, Russia, and Eastern Mediterranean. PMCAMx-UF predicted a 10% increase in $N_{100}$ in the United Kingdom (Fig. S3).

The increase in the biogenic nucleation rate by a factor of 10 in Case 5 resulted in a significant increase of 150-200% for the $N_{tot}$ (15000-20000 cm$^{-3}$) in the areas with intense nucleation, and a 50-70% increase in $N_{10}$ (over 3000 cm$^{-3}$) in Western Europe, Turkey, and Scandinavia (Fig. S4). In the case of $N_{50}$, there was an increase of about 15-20% in the regions of Scandinavia and Northern Russia, and 10-15% in the eastern Mediterranean. For $N_{100}$ there was a small increase for almost all the domain with a peak change of 5%-8% in the Balkans and Turkey (Fig. S4).

The reduction by a factor of 10 in the nucleation rate in Case 6 led to a 50-70% reduction in $N_{tot}$ (5000-7000 cm$^{-3}$) and a 35-50% reduction in $N_{10}$ (2500-3500 cm$^{-3}$) for the entire simulated area (Fig. S5). For $N_{50}$ there was a decrease of about 20-25% in Scandinavia and Northern Russia and about a 15-20% reduction in the eastern Mediterranean. In the case of $N_{100}$ there was a small decrease of 5%-8% in the eastern Mediterranean Sea (Fig. S5).

### 4.3.2 Effect initial nuclei diameter in the biogenic nucleation parameterization

The reduction of the nuclei diameter from 1.7 nm to 1 nm in Case 7 resulted in a 25-35% reduction of $N_{tot}$ (2500-3500 cm$^{-3}$) and a 20-25% decrease in $N_{10}$ (1500-2000 cm$^{-3}$) in the Balkans, Poland, and Russia where intense nucleation events were predicted (Fig. S6). For $N_{50}$ and $N_{100}$, a reduction of about 5% is predicted. The reduction of the nuclei diameter mainly affects the number of particles between 1-10 nm. The smaller initial diameter leads to an acceleration of coagulation and leads to
faster losses of those fresh particles. For this reason, a significant reduction in $N_{tot}$ is predicted in the eastern Mediterranean Sea and the Balkans, where the highest concentrations of the largest ($N_{50}$ and $N_{100}$) particles are found.

### 4.3.3 Effect of ELVOCs in nucleation

In this case the semi-volatile biogenic organics (C*=1 μg m$^{-3}$) were substituted by the biogenic ELVOCs (C*=10$^{-5}$ μg m$^{-3}$) in
the parameterization. This was accompanied by an increase in the scaling factor from 10$^{-22}$ to 10$^{-21}$ molecule$^{-3}$ cm$^6$ s$^{-1}$. This modification resulted in a predicted increase of 40-100% in $N_{tot}$ (2000-4000 cm$^{-3}$) and a 10-40% increase for $N_{10}$ (500-2000 cm$^{-3}$) compared to the base bSOA parameterization, across regions including Portugal, northern France, the United Kingdom, Germany, Poland, southern Scandinavia, the Balkans, and Russia. Conversely, a reduction of approximately 30% in $N_{tot}$ and 20% in $N_{10}$ is predicted for the Mediterranean region (Fig. 6). For $N_{50}$ an increase of 5-10% (100-200 cm$^{-3}$) was predicted in



Poland and Scandinavia, while a slight decrease of 5% is shown for the Mediterranean Sea. The change in $N_{100}$ was less than 10%, with the most significant differences occurring in Portugal, Turkey, Scandinavia, and the United Kingdom.

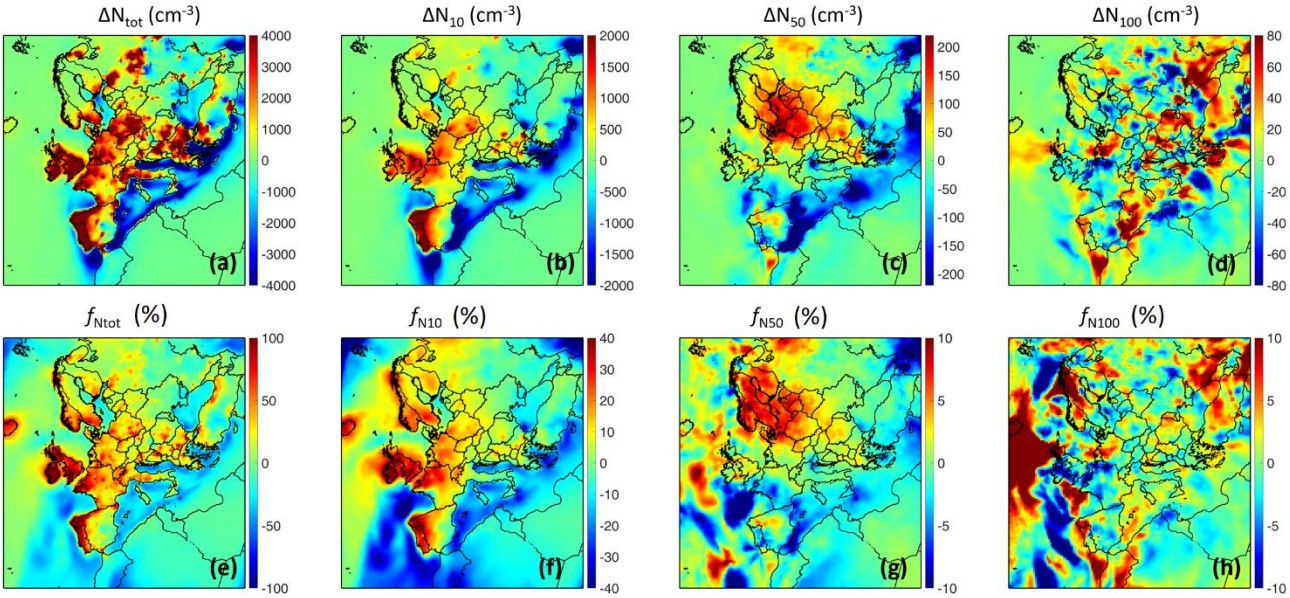

**Figure 6: Average ground increase of number concentration (in cm⁻³) (a-b-c-d) and fractional increase ($f_{Nx}$) of number concentration (in %) (e-f-g-h) for case 8 (ELVOCs as third species) of organic nucleation during 5 June – 8 July 2012 for: (a-e) all particles ($f_{Ntot}$); (b-f) particles above 10 nm ($f_{N10}$); (c-g) above 50 nm ($f_{N50}$); and (d-h) above 100 nm ($f_{N100}$). Different scales are used.**

For the case of the sulfuric-acid ELVOCs nucleation, high nucleation rates are predicted in the United Kingdom, Portugal, northern Spain, northern Italy, Poland, the Balkans, Turkey, and Russia (Fig. 7c). In these areas there are both high concentrations of ELVOCs and sulfuric acid according to PMCAMx-UF (Fig. 7a, b).

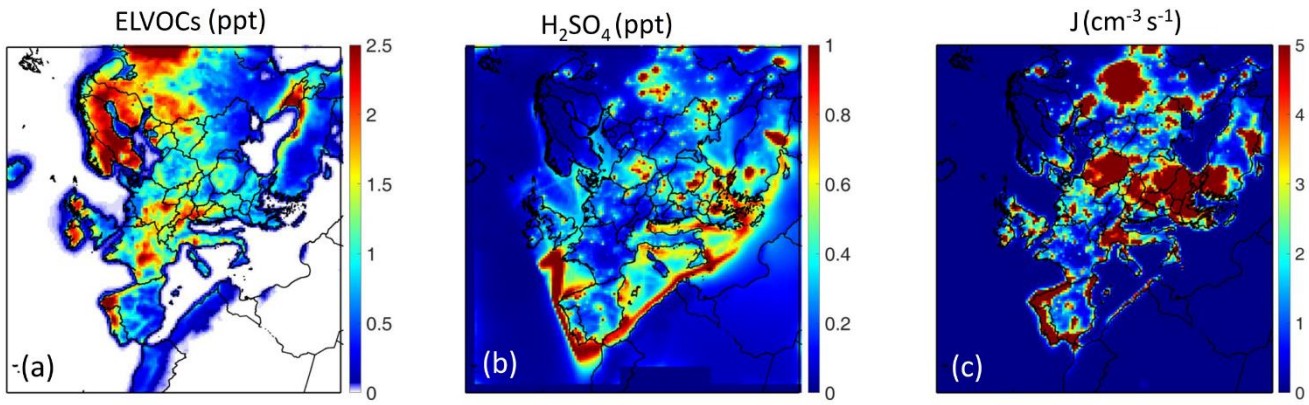

**Figure 7: Ground level average gas concentration for case 8 of (a) extremely low-volatility organic compounds (ELVOCs) with $C^*=10^{-5}$ µg m⁻³ (in ppt) and (b) sulfuric acid (in ppt) and (c) nucleation rate $J$ (in cm⁻³ s⁻¹) for the organic nucleation during 5 June-8 July. Different scales are used.**




## 4.4       Evaluation of all simulation cases

A scenario excluding nucleation has been included in the evaluation for comparative purposes. This no-nucleation scenario significantly underestimates $N_{10}$ concentrations, whereas the incorporation of nucleation improves significantly model predictions across all investigated cases (Cases 1-8; Fig. 8). All simulations with nucleation result in predicted distributions of $N_{10}$ concentrations that are consistent with the observed measurement range. The exception is the scaled up biogenic-sulfuric acid parameterization (Case 5) that overpredicts the $N_{10}$ concentrations in a lot of the stations. The median observed

concentration of $N_{10}$ is close to Cases 1 and 2, both of which employ ammonia as a third species, but also Cases 4, 7 and 8, which are based on biogenic organic vapors.

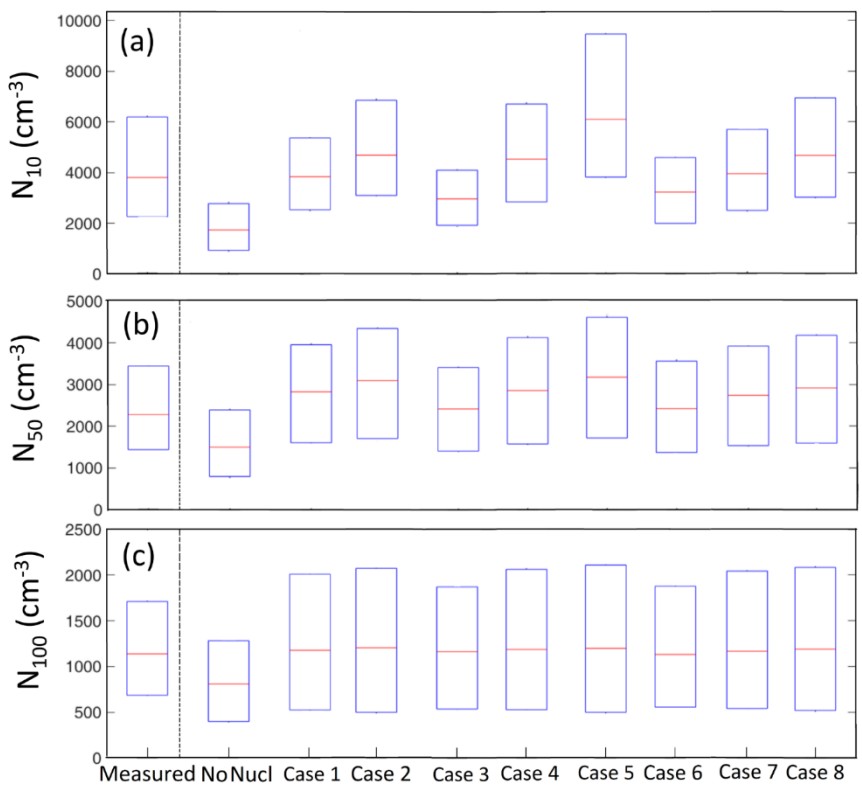

**Figure 8: Measurements from 26 ground stations, against the simulation without nucleation, the ammonia ternary parameterization (Case 1) and the change by an order of magnitude in scaling factor (Cases 2 and 3), the biogenic parameterization (Case 4) with the**

**change by an order of magnitude in scaling factor (Cases 5 and 6), decrease of nuclei diameter (Case 7) and the ELVOCs addition as the third species (Case 8) for a) $N_{10}$, b) $N_{50}$ and c) $N_{100}$. The lower and upper lines in each box represent the 25% and 75% of the results respectively, while the middle line corresponds to the median value.**

The $N_{50}$ concentrations are clearly underestimated in the no nucleation simulation. In both scenarios in which the

nucleation rate was reduced by an order of magnitude (Cases 3 and 6), the predicted $N_{50}$ concentration is closer to the



measurements both in in terms of median $N_{50}$ and the range of values (Fig. 8b). The remaining Cases (1, 2, 4, 5, 7 and 8) overestimate the median $N_{50}$; however, the corresponding ranges of values are close to the measurements.

In the case of $N_{100}$, the no nucleation case significantly underestimates its concentration. Conversely, in all nucleation tests, the predicted median $N_{100}$ concentration is close to the measurement values. At the same time PMCAMx-UF predicts a
broader range of $N_{100}$ values for Cases 1-8 in relation to the measurements (Fig. 8c).

The no nucleation simulation underestimates all number concentrations with a NMB of -60% for $N_{10}$, a NMB of – 30% for $N_{50}$ and a NMB of -27% for $N_{100}$ (Fig. 9). Cases 1, 2, 4, 7, and 8 exhibit a NMB of ±20% for $N_{10}$, Cases 3 and 6 (both involving a tenfold reduction in nucleation rate) show a NMB between –30% and -40% (Fig. 9a). Case 5, which involves bSOA and an increased nucleation factor, has the highest NMB of all, at 50%.

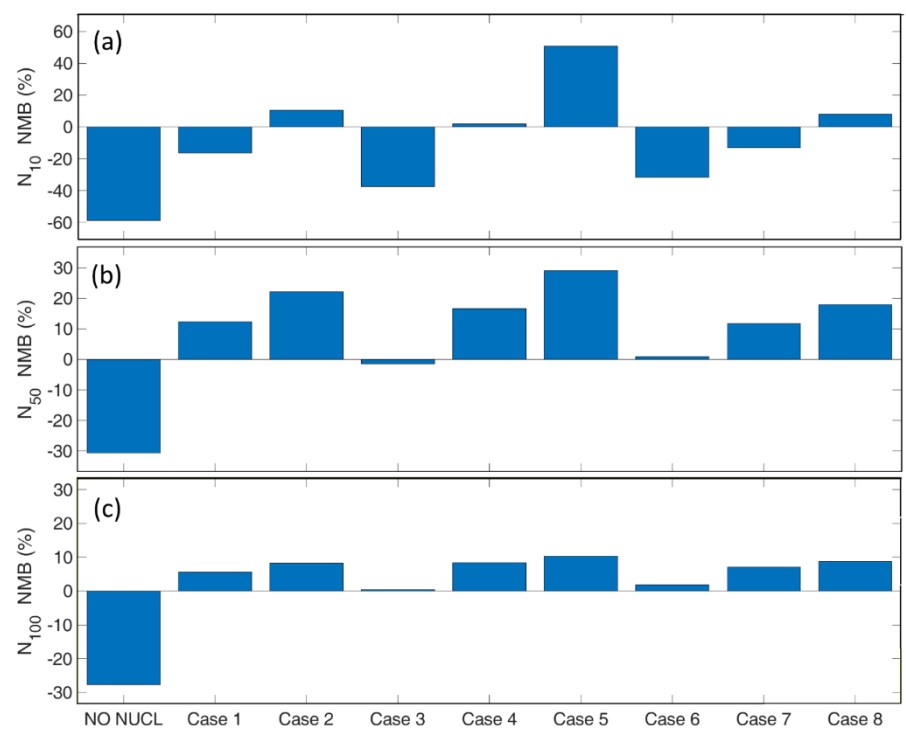


**Figure 9: The NMB for hourly a) N₁₀, b) N₅₀ and c) N₁₀₀ for the no-nucleation scenario, the ammonia ternary parameterization (Case 1) and the change by an order of magnitude in scaling factor (Cases 2 and 3), the biogenic parameterization (Case 4) with the change by an order of magnitude in scaling factor (Cases 5 and 6), decrease of nuclei diameter (Case 7) and the ELVOCs addition as the third species (Case 8).**


For $N_{50}$, the simulations in which the nucleation rate was reduced by a factor of 10 exhibit the lowest NMB which was close to zero. The cases where the nucleation rate was increased by 10 times (Cases 2 and 5) presented the maximum NMB among all the simulated scenarios with a NMB of 22% and 30%, respectively (Fig. 3b). For the rest of the Cases (1,4, 7 and 8), the NMB varies between 0 and 20% (Fig. 9b).

For N$_{100}$, all cases incorporating ammonia or bSOA nucleation exhibit a NMB of less than 10%. The cases in which
the nucleation rate was reduced by an order of magnitude (Cases 3 and 6) demonstrate the lowest NMB which was close to
zero (Fig.9c).

The normalized mean error (NME) for $N_{10}$ ranges between 50 and 60 % for nearly all examined parameterizations
with the only exception being Case 5 (biogenic and increased scenario) for which NME exceeds 80% (Fig. S7a). For $N_{50}$ the
lowest NME was found for the reduced scaling factor for both ammonia (Case 3) and biogenic (Case 6) parameterization (Fig.
S7b). Regarding $N_{100}$, all cases presented an NME of less than 50% (Fig. S7c).

Soccer plots, which depict fractional bias as a function of fractional error, are utilized to illustrate model performance
(Morris et al., 2005). In Fig. 10, the performance of PMCAMx-UF is shown for the examined parameterizations and for all
measurements in all stations using daily temporal resolution. For the no nucleation scenario, the model performance for $N_{50}$
and $N_{100}$ was average (F$_{bias}$ < ±60% and F$_{error}$ < ±75%). However, for the $N_{10}$, the performance fell outside this range, indicating
fundamental errors and underscoring the necessity of incorporating nucleation processes for accurate $N_{10}$ prediction. For $N_{10}$,
the ammonia (Case 1) and biogenic (Case 4) parameterization, along with the nuclei size adjustment (Case 7) and the use of
ELVOCs (Case 8), show good performance. The scenarios involving scaling factor adjustments (either increased or decreased
by an order of magnitude) border on the good and excellent performance regions (Cases 2, 3, 5, and 6). For $N_{50}$ and $N_{100}$, all
eight investigated parameterizations have good performance (F$_{bias}$ < ±30% and F$_{error}$ < ±50%) and are very close to the criteria
for excellent performance (< ±30% and F$_{error}$ < ±50%).

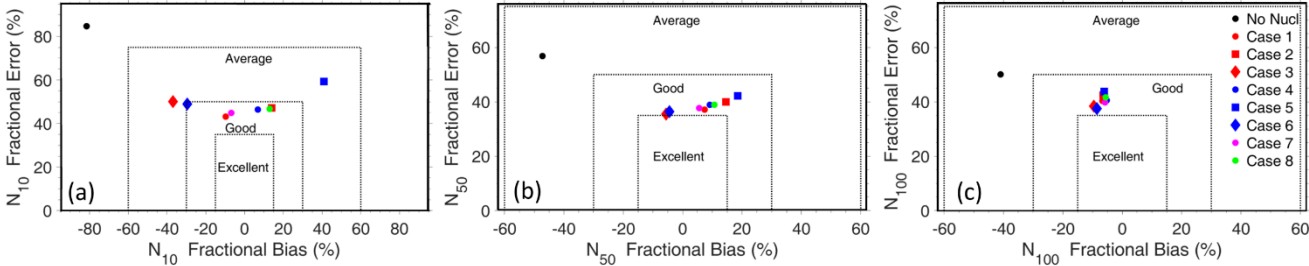

**Figure 10: Model evaluation using fractional error (%) versus fractional bias (%) of daily number concentrations for (a) $N_{10}$, (b) $N_{50}$**
**and (c) $N_{100}$ for the no-nucleation scenario, the ammonia ternary parameterization (Case 1) and the change by an order of magnitude
in scaling factor (Cases 2 and 3), the biogenic parameterization (Case 4) with the change by an order of magnitude in scaling factor
(Cases 5 and 6), decrease of nuclei diameter (Case 7) and the ELVOCs addition as the third species (Case 8).**

The performance of PMCAMx-UF for various cases was also analysed using the soccer plots for each one of the 26
sites across Europe using once more daily temporal resolution (Fig. S8). The PMCAMx-UF performance for $N_{100}$ for most
stations is good or excellent, for both the ammonia and biogenic organic nucleation cases. For $N_{10}$, the ammonia (Case 1)
parameterization performs a little better than the biogenic cases (4 and 8).





## 4 Conclusions

In this study, we considered two nucleation parameterizations involving sulfuric acid and water: one in which ammonia was the third reactant and one in which semi-volatile biogenic organics participated in the critical cluster. The parameters of both expressions were selected so that the predicted rates would be generally consistent with available ambient nucleation rate measurements. Nucleation enhanced the $N_{tot}$ by 160-300%, the $N_{10}$ 140-180% and the $N_{100}$ 45-50% during the simulated period. The base case organic parameterization, when implemented in PMCAMx-UF tended to predict higher $N_{10}$ concentrations over

Europe that were, on average, 40-60% higher compared to the ammonia case. This is a relatively small difference given the substantial differences between the two nucleation mechanisms. The biogenic organic parameterization predicted values of $N_{10}$ that were 30-50% higher over the Mediterranean, more than 50% higher in Russia, and 20% higher in Scandinavia compared to the ammonia parameterization predictions. There were a few areas in central and western Europe in which the opposite was true, with 20% lower $N_{10}$ values predicted when the biogenic organic parameterization was used.

420        Despite the significant differences in the used parameterizations, the average predicted $N_{100}$ concentrations over the domain differed by less than 5%. This suggests surprisingly low sensitivity of the current concentrations of these larger particles (a proxy for CCN) to the details of the nucleation mechanism, provided the parameterizations are consistent with the available ambient observation dataset.

        Both parameterizations demonstrated good performance against hourly measurements at 26 stations, with similar

accuracy. The simulation with ternary ammonia nucleation had a NMB for $N_{10}$ of −16% and for $N_{100}$ equal to 6%. The performance for the biogenic organic parameterization had a lower NMB of 2% for $N_{10}$, but a little higher (8%) for $N_{100}$. The NME in $N_{10}$ for both simulations was below 60% for most of the stations and was quite similar for the two parameterizations. Modifying the ammonia nucleation rate parameterizations by an order of magnitude led to average changes in predicted $N_{10}$ concentrations by ±30% and $N_{100}$ by -5% to 2%. Similar adjustments in biogenic aerosol nucleation rates resulted in average

changes of -30% to 40% for $N_{10}$ and -5% to 2% for $N_{100}$. Decreasing the nuclei diameter for biogenic organic nucleation from 1.7 nm to 1 nm caused a significant decrease in $N_{10}$, particularly over the Mediterranean Sea and central Europe, with average changes of -20%. Incorporating ELVOCs as a third species resulted in an average change of 3% in $N_{10}$ and 0.4% in $N_{100}$, aligning well with observed number concentrations at most stations. These adjustments represent relatively modest differences given the divergent nucleation mechanisms involved.

*Author contributions*

**DP:** writing – original draft, writing – review & editing, methodology, investigation, formal analysis, conceptualization. **KF:** writing – review & editing. SNP: writing – original draft, writing – review & editing, supervision, project administration, methodology, investigation, conceptualization.



*Declaration of competing interest*

The authors declare that they have no known competing financial interests or personal relationships that could have appeared to influence the work reported in this paper.

*Code and data availability*

The model code base used to generate the results for ammonia ternary nucleation (PMCAMx-UF version 2.1) can be found on Zenodo at https://zenodo.org/records/10078189 (Patoulias, D., & Pandis, S. 2024a). The model code base used to generate

the results for biogenic nucleation (PMCAMx-UF version 2.2) can be found on Zenodo at https://zenodo.org/records/12720811 (Patoulias, D., & Pandis, S. 2024b). The analysis codes and data used to prepare the manuscript can be found on Zenodo at https://zenodo.org/records/13348332 (Patoulias et al., 2024c).

*Acknowledgements*

This work was supported by the Atmospheric nanoparticles, air quality and human health (NANOSOMs) project funded by

the Hellenic Foundation for Research & Innovation (HFRI) (grant agreement no. 11504) and the RI-URBANS project (grant agreement No 101036245 from the European Union's Horizon 2020 research and innovation program).

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
