# Peer review of "Sensitivity of predicted ultrafine particle size distributions in Europe to different nucleation rate parameterizations using PMCAMx-UF v2.2"

_Geoscientific Model Development, 2024_

## Author Response (AR1)

**Responses to the Comments of the Reviewers**

**Reviewer 1**

**(1)** This paper compares ultrafine aerosol concentrations in Europe in PMCAMx simulations with two different new particle formation mechanisms. The base case is a scaled ternary NPF parameterization by Napari et al (2002) involving $H_2SO_4$ and $NH_3$. The adjusted case is an organic-inorganic parameterization by Riccobono et al (2014). The base case and model used was previously evaluated in much the same way over the same domain by Patoulias et al (2018). A key finding is that, on the land surface in Europe, the predicted 100 nm-particle concentrations do not depend strongly on the particle formation mechanism, at least provided that they are in good agreement with observations. While the study does not describe ambitious model developments or innovative evaluations compared to previous work by the same authors, the detailed, dedicated study of the sensitivity to simple new particle formation parameterizations is useful and the well-motivated tests were interesting and in some cases novel, for example, the sensitivity to the reduction of the nucleation diameter. The method of adjustment of the particle formation parameterizations to suit the species the model represents is also novel and interesting. The model is state-of-the-art, and the evaluation is thorough for the measurements used, although these are limited to ground sites. I think the paper could be suitable for publication in GMD if the comments below can be addressed.

We would like to thank the reviewer for the positive assessment and the constructive comments. Our responses and the corresponding changes (in black font) follow each comment (in blue).

**Scientific comments**

**(2)** A key message of the paper is that the detailed mechanism for particle formation does not strongly affect the simulated particle number concentrations. Once this is understood, it then follows that it is not critical that the particle formation mechanisms discussed in the paper are state of the art. Still, I did not understand why the parameterization of Baranizadeh et al. was not also used. Perhaps the code is no longer available, but some explanation would be good (even if the answer is not very exciting). Also, other relatively similar but more up-to-date options for inorganic particle formation are now available, for example the parameterization of Yu et al., who presents lookup tables in a GMD 2020 paper. This parameterization includes the potentially important contribution of ions. Similarly, a parameterization of organic-inorganic NPF is available in Lehtipalo et al., Science Advances (2018) which has some similar limitations to Riccobono et al. (2014), such as being valid at only one temperature, but at least it is based on chamber experiments in which ELVOC were detected by a mass spectrometer rather than inferred imprecisely. Some discussion of this more recent work might still be useful in the manuscript.

The parameterization proposed by Baranizadeh et al. (2016) is still available in PMCAMx-UF; however, it tended to overpredicted concentrations for particles with diameters between 10 and 100 nm compared to the scaled parameterization by Napari et al. (2002). Therefore, the authors

chose the latter as the basis for their analysis of sulfuric acid-ammonia-water nucleation parameterizations. Our goal in this study was to compare one parameterization based on ammonia and one on organics and try to dry conclusions from this comparison. A brief discussion of this choice together with references to the studies by Yu et al. (2020) and Lehtipalo et al. (2018), that could have been used, have been added to the revised manuscript.

**(3)** It makes sense to adjust the rate constant given in Riccobono et al, which is very uncertain as well as not representative of the VBS species in PMCAMx. However, it took me a little time to understand how Chen et al. is being used (simply to bound the range of possible nucleation rates for a given sulfuric acid concentration, such that to a first approximation it does not matter which species are forming particles). In Chen et al. (2012), the participation of ELVOCs in particle formation might be very different to the current study. Some more discussion of how this might influence the results would be useful. The caption of Figure S1 would ideally say where exactly the data in the plot come from (I think the PMCAMx simulation, not measurements, right?) I think the plot needs a bit more explanation because (if I understand correctly) it's reversed compared to almost all data analysis anyone ever sees, in that the "fit line" is used to get the individual data points, rather than the other way around. Also, it seems important enough to the study to put it into the main text.

We have added the corresponding explanation of what exactly is shown in Figure S1 and the origin of the corresponding data. Indeed, the data shown corresponds to PMCAMx-UF simulation data. We also explain that the Chen et al. (2012) analysis is used just to bound the nucleation observations, and their bounds are adopted here and are shown in Figure S1. This information has been added to the revised paper.

**(4)** Some discussion of the size dependence of condensational growth rates, and coagulation rates, in the 1-1.7 nm range and its effect would be interesting and welcome in section 4.3.2.

A brief discussion of the condensational growth and coagulation rates has been added in section 4.3.2 together with a reference to Pierce and Adams (ACP, 2007) who have presented a detailed analysis of the interplay of these processes.

**(5)** The NMB results quoted in the conclusion, at lines 424-425, look too good to be true, at first glance. It seems like a different metric might give a more realistic summary of the results (NME?) Looking at the conclusion left me wondering what kind of overfitting or tuning had been done to get NMB values so low given (for example) the large uncertainties in ammonia, BVOC and $SO_2$ emissions that the particle formation and growth rates depend on, and the sparsity of measurements of their concentrations. Then Figures 4 and 5 reassured me that the biases were still more or less as expected from this kind of work at individual stations. As the model underpredicts at some stations and overpredicts at others, it is presumably partly luck that the stations are distributed such that the biases end up cancelling out in the average. Some more discussion of this, and whether or not the results are right for the right reasons, might be useful. Even though a clear

message of the paper is that the fine details of the new particle formation mechanism don't matter very much to the particle number concentrations.

We agree with the reviewer that the low NMB is partially due to the fact that the model overpredicts in some stations and underpredicts in others. This point is discussed in the revised paper together with the corresponding error metric.

**(6)** It would be great to investigate whether the key result, the lack of sensitivity to particle formation mechanism, generalizes to higher altitudes. Even if no evaluation can be done, the particle concentration vertical profiles might still be interesting. In the conclusion (and abstract), should there also be a caveat that the lack of sensitivity to particle formation mechanism may not generalize outside Europe, e.g. to tropical rainforests, deserts, pristine oceans, or extensive ice sheets?

The PEGASOS data set also includes measurements aloft by a Zeppelin. The predicted vertical profiles predicted by the two parameterizations over the Po Valley were quite similar and in good agreement with the parameterizations. A brief discussion of this point has been added to the paper together with a figure showing the comparison in the supplementary information. The caveat suggested by the reviewer is reasonable and has been added to the abstract and the conclusions.

**Minor technical or editorial comments**

**(7)** Was Riccobono et al added to the Napari parameterization or simply used in its place? I assume the latter, from later context, but it would be good to be explicit around line 140.

The Riccobono et al. (2014) parameterization was used instead of the Napari et al. (2002) parameterization in our simulations. While it is possible to include both parameterizations, this will be explored in future work. This clarification has been added to the main text.

**(8)** What was the time frequency of simulation output at the EUCAARI sites? Is it instantaneous output or means?

The time frequency of the output of PMCAMx-UF is selected by its user. An hourly output has been used in this study. This is now stated in the paper.

**(9)** Typo in title of section 4.3.2

We have corrected the typo.

**(10)** Abstract line 20: "Among the tested scenarios, the measurements showed better agreement with the ternary ammonia and ELVOC-based parameterizations for $N_{10}$"—compared to what? A scenario with no NPF?

These two parameterizations are in better agreement with the field data. The sentence has been rephrased for better clarity.

**Reviewer 2**

**(1)** The manuscript presents an extensive analysis of the sensitivity of two nucleation parameterizations. According to the authors, despite differences in details of the simulations and results, the bottom line is that their results suggest "low sensitivity of ... larger particles (a proxy for CCN) to the details of the nucleation mechanism ...". I think this is a reassuring result for the modeling community: Increased complexity may not be as rewarding as anticipated. Overall, while not flashy, I think this is an excellent manuscript and I fully recommend it for publication as is. The manuscript is well written, well presented, and well argued. Given Reviewer 1 covers most of my other comments, I will make mine brief. I encourage the authors to reflect on the second paragraph of their Conclusions section (which Reviewer 1 and I found interesting). It would be interesting to read more of your thoughts on the implications of this low sensitivity and where we go from here. Below I list a few minor comments that the reviewers are welcome to respond to and address in the revised manuscript, but these are very minor, and it is okay to ignore them. They are mainly about the vertical coordinate in the model used.

We would like to thank the reviewer for the positive assessment and the comments. Our responses and the corresponding changes (in black font) follow each comment (in blue).

**(2)** S3: Model Application. To be sure, when you say 36x36 km grid, these are perfect squares on a structured grid, right?

Yes, this is true.

**(3)** Could you describe the 14 vertical layers in more detail? Are they uniform? Is the vertical coordinate terrain following or not?

The 14 vertical layers, including their non-uniform structure and exact bottom and top heights, are now provided in the supplementary information. The vertical coordinate is terrain following.

**(4)** Clarify what you meant by "A rotated polar stereographic map projection was used for the simulations by PMCAMx-UF"

A rotated polar stereographic map projection was employed in the simulations with PMCAMx-UF to focus on Europe as the primary area of interest. This projection minimizes distortions in spatial representation across the region by aligning the projection's central point with the area of study. The rotation ensures that Europe is accurately represented while maintaining consistency in horizontal grid spacing, which is essential for atmospheric modeling. A reference to the CAMx manual has been added for the interested reader.

**(5)** How were the meteorological inputs used to derive the simulation? They prescribe the fields at each time step? What's the time step?

The meteorological inputs (including temperature, pressure, horizontal wind components, water vapor, vertical diffusivity, clouds, and rainfall) were generated by the Weather Research and

Forecasting (WRF) model (Skamarock et al., 2005). The meteorological inputs correspond to hourly data. This information has been added to the paper.

**(6)** How were the vertical layers of WRF and PMCAMx-UF were unified? Interpolation? Or did you choose the model coordinate to be the same for both?
The vertical layers of WRF and PMCAMx-UF use the same coordinates and heights. This is also mentioned in the revised paper.

(7) **S4: Results.** Maybe I missed it, but how did you define ground level? Grid-mean in the first layer? Or at the bottom interface of the first layer? What's the height of the first layer?
The first layer corresponds to a bottom height of 0 m (surface) to a top height of 60 m. The ground level is defined as this first layer, meaning the first 60 m. The 14 vertical layers, including their exact bottom and top heights, are now included in the Supplementary Information.

**References**

Baranizadeh, E., Murphy, N. B., Julin, J., Falahat, S., Reddington, L. C., Arola, A., Ahlm, L., Mikkonen, S., Fountoukis, C., Patoulias, D., Minikin, A., Hamburger, T., Laaksonen, A., Pandis, N. S., Vehkamäki, H., Lehtinen, E. J. K. and Riipinen, I.: Implementation of state-of-the-art ternary new-particle formation scheme to the regional chemical transport model PMCAMx-UF in Europe, Geosci. Model Dev., 9, 2741–2754, doi:10.5194/gmd-9-2741-2016, 2016.

Lehtipalo, K., Yan, C., Dada, L., Bianchi, F., Xiao, M., Wagner, R., Stolzenburg, D., Ahonen, L.R., Amorim, A., Baccarini, A., Bauer, P.S., Baumgartner, B., Bergen, A., Bernhammer, A.-K., Breitenlechner, M., Brilke, S., Buchholz, A., Mazon, S.B., Chen, D., Chen, X., Dias, A., Dommen, J., Draper, D.C., Duplissy, J., Ehn, M., Finkenzeller, H., Fischer, L., Frege, C., Fuchs, C., Garmash, O., Gordon, H., Hakala, J., He, X., Heikkinen, L., Heinritzi, M., Helm, J.C., Hofbauer, V., Hoyle, C.R., Jokinen, T., Kangasluoma, J., Kerminen, V.-M., Kim, C., Kirkby, J., Kontkanen, J., Kürten, A., Lawler, M.J., Mai, H., Mathot, S., Mauldin, R.L., Molteni, U., Nichman, L., Nie, W., Nieminen, T., Ojdanic, A., Onnela, A., Passananti, M., Pet¨aj¨a, T., Piel, F., Pospisilova, V., Qu´el´ever, L.L.J., Rissanen, M.P., Rose, C., Sarnela, N., Schallhart, S., Schuchmann, S., Sengupta, K., Simon, M., Sipil¨a, M., Tauber, C., Tom´e, A., Tr¨ostl, J., V¨ais¨anen, O., Vogel, A.L., Volkamer, R., Wagner, A.C., Wang, M., Weitz, L., Wimmer, D., Ye, P., Ylisirni¨o, A., Zha, Q., Carslaw, K.S., Curtius, J., Donahue, N.M., Flagan, R.C., Hansel, A., Riipinen, I., Virtanen, A., Winkler, P.M., Baltensperger, U., Kulmala, M., Worsnop, D.R.: Multicomponent new particle formation from sulfuric acid, ammonia, and biogenic vapors, Sci. Adv., 4, eaau5363, https://doi.org/10.1126/sciadv.aau5363, 2018.

Napari, I., Noppel, M., Vehkamäki, H. and Kulmala, M.: Parametrization of ternary nucleation rates for $H_2O$-$H_2SO_4$-$NH_3$ vapors, J. Geophys. Res. Atmos., 107, 4381, doi:10.1029/2002JD002132, 2002.

Patoulias, D., Fountoukis, C., Riipinen, I., Asmi, A., Kulmala, M. and Pandis, S. N.: Simulation of the size-composition distribution of atmospheric nanoparticles over Europe, Atmos. Chem. Phys., 18, 13639–13654, doi:10.5194/acp-18-13639-2018, 2018.

Skamarock, W. C., Klemp, J. B., Dudhia, J., Gill, D. O., Barker, D. M., Wang, W. and Powers, J. G.: A description of the advanced research WRF version 2, 100, doi:10.5065/D6DZ069T, 2005.

Yu, F., Nadykto, A. B., Luo, G., and Herb, J.: $H_2SO_4$–$H_2O$ binary and $H_2SO_4$–$H_2O$–$NH_3$ ternary homogeneous and ion-mediated nucleation: lookup tables version 1.0 for 3-D modeling application, Geosci. Model Dev., 13, 2663–2670, https://doi.org/10.5194/gmd-13-2663-2020, 2020.